# Modeling Contact Stiffness of Soft Fingertips for Grasping Applications

**DOI:** 10.3390/biomimetics8050398

**Published:** 2023-09-01

**Authors:** Xiaolong Ma, Lingfeng Chen, Yanfeng Gao, Daliang Liu, Binrui Wang

**Affiliations:** 1College of Mechanical and Electrical Engineering, China Jiliang University, Hangzhou 310018, China; mxlslf@cjlu.edu.cn (X.M.); gaoyanfeng@cjlu.edu.cn (Y.G.); 2The Key Laboratory of Special Purpose Equipment and Advanced Processing Technology, College of Mechanical Engineering, Ministry of Education and Zhejiang Province, Zhejiang University of Technology, Hangzhou 310023, China; 1111802011@zjut.edu.cn; 3He’nan Jiuyu Boda Industrial Co., Ltd., Zhengzhou 450051, China; 06a0103088@cjlu.edu.cn

**Keywords:** soft fingertip, contact stiffness, elastic model, grasp stability

## Abstract

Soft fingertips have distinct intrinsic features that allow robotic hands to offer adjustable and manageable stiffness for grasping. The stability of the grasp is determined by the contact stiffness between the soft fingertip and the object. Within this work, we proposed a line vector representation method based on the Winkler Model and investigated the contact stiffness between soft fingertips and objects to achieve control over the gripping force and fingertip displacement of the gripper without the need for sensors integrated in the fingertip. First, we derived the stiffness matrix of the soft fingertip, analyzed the contact stiffness, and constructed the global stiffness matrix; then, we established the grasp stiffness matrix based on the contact stiffness model, allowing for the analysis and evaluation of the soft fingertip’s manipulating process. Finally, our experiment demonstrated that the variation in object orientation caused by external forces can indicate the contact force status between the fingertip and the object. This contact force status is determined by the contact stiffness. The position error between the theoretical work and tested data was less than 9%, and the angle error was less than 5.58%. The comparison between the theoretical contact stiffness and the experimental results at the interface indicate that the present model for the contact stiffness is appropriate and the theoretical contact stiffness is consistent with the experiment data.

## 1. Introduction

Soft robotics has gained significant attention in recent years due to its advantages over traditional rigid robots [1,2]. Soft robotic fingers [3], grippers [4], and humanoid hands have been developed to provide flexible and compliant contacting and manipulating capabilities [5]. However, despite the numerous studies on sophisticated designs and manipulations, soft hands are far from suitable for commercial applications in industrial service manipulations. When performing skillful manipulation tasks, soft robotics are essential for achieving the correct position and attitude of the object, which poses a challenge for the sensorimotor control system. Technical challenges still exist in both modeling and control strategies to make soft fingers as dexterous as those of humans.

When a soft finger comes into contact with a rigid object, multiple three-dimensional contact elements exert force on the contact surfaces. Researchers have discovered that the contact force and displacement exhibit strong nonlinearity, and the friction coefficient may change with varying contact conditions. Additionally, as the tangential forces in the tangential directions increase, the contact nodes may separate or re-establish contact, leading to a variable contact stiffness. In this paper, we assume that the contact elements are capable of supporting a compressive load in the normal direction and tangential forces in tangential directions. The displacements in the tangential and normal directions are assumed to be independent; the parameters can be equivalent to three independent contact springs: two with stiffness *k_t_* along tangential direction; one with stiffness *k_n_* along normal direction. For a long time, the modeling of contact stiffness in soft robotic hands has been fundamental to object grasping and manipulating; it is impossible or extremely difficult to establish a theoretical model of contact stiffness on the grasp surface between soft fingertips and an object, and the dynamic properties of the grasp system are generally inaccurately predicted. Most soft multiple fingers use an adaptive open-loop control method for grasping [6,7]. While most researchers have focused on the contact point between the manipulator and the object [8,9], few have considered the friction issue upon the contact point [10]. The research on contact constraint is still insufficient, and current torque constraint investigations mainly focus on rigid contact assumptions, ignoring the elastic process during grasping [11,12].

One of the most common methods for modeling arbitrary surface contact is to use finite element models [13,14] or half-space models [15,16]. Although finite element models [17] have been used to simulate contact on elastic surfaces and can be easily extended to elastoplastic behavior, they are limited in their application to soft finger grasping and manipulation due to the large quantity of unknowns and the corresponding computational time. Half-space models require less computational time, however, because they are based on analytical solutions for the elastic case, i.e., they cannot be simply extended to the state of elastoplastic deformation. Based on the FEA model, contact conditions are classified into the following three cases: open condition; stick condition; sliding condition. In this paper, we are addressing the stick condition: the gap remains closed, and no sliding motion occurs in the tangential directions. Researchers have investigated the issue of stiffness and flexibility for robotics by employing matrices and eigenvalues to represent contact compliance characteristics and reveal the stiffness matrix’s invariance [18]. A stiffness model [19,20] was developed with Lie algebra to reveal the spring-like constraint characteristics of rigid contacting, and a new definition of grasping and manipulating forces for multi-fingered robot hands was proposed [21]. However, existing soft fingertip models, such as the power-law model [22], the parallel distributed model [23], and the radially distributed model [24], are not easy to calculate, especially for real-time manipulation.

To address this issue, an elastic constraint model for grasping was established under both friction and frictionless conditions [25]; it presents the analysis of restraint of a rigid body in contact with any arrangement of point contacts with known frictional properties. The relationship is established between externally applied loads and contact forces in the friction condition. mathematical models were found for the frictional constraints using convex set theory. Ref. [26] presents a geometrical representation and theoretical foundation of robot grasping that is affected by friction and by the magnitudes of normal contact forces. The constraint cone and the constraint region describe the effect of friction and the influence of the amplitudes of normal forces. Based on these theories, researchers investigated the finite displacements of rigid bodies under different contact force distributions during manipulation with a group of preloaded springs [26], discussing the form-closure robotic grasping of a rigid object and showing how fine motion of the grasped object may be induced. The manipulator’s contact stiffness was defined to examine soft contact mechanisms and achieve dexterous manipulation [27]. Indeterminate grasping systems were solved by formulating contact stiffness and employing geometric compatibility models [28,29], describing the grasping of an arbitrary planar object using frictionless point contacts. The grasping configuration was proven to have a significant influence on the stiffness matrix [30]. Two types of stiffness matrices are discussed: constant and configuration-dependent matrices. The constant-stiffness matrix can be derived from a conservative quadratic potential function in the Hermitian form; while the skew-symmetric part is a function of the nonconservative curl vector field of the grasp. Experiments showed that the finite displacement produced by a group of preloaded springs on a rigid manipulator has an evident effect on grasping stiffness [31,32]; the forces are provided as preloads or the effect of preloads which can be applied through two types of pre-loading configurations, one of which is to apply a preload parallel to a contact normal and the other is through the contact normal. The Winkler Model is a dimension reduction method that can be used to map a three-dimensional contact problem to one dimension [33]. This model consists of a set of linear units with very small spacing and with each set as a linear spring, simplifying theoretical analysis and numerical calculation. Despite efforts and achievements in this area, the contact stiffness of soft fingertips remains in the investigation stage. Inspired by the Winkler Model, we propose a novel soft fingertip contact model that is promising for implementation in soft hands grasping and manipulating.

The contributions of this work are as follows: (1) the force diagram at an individual elastic contact point can be represented as three springs, and the contact stiffness matrix can be established with screw algebra; (2) we developed a representation of soft fingertips at the contact surface to analyze the contact stiffness matrix of soft fingertips; (3) we present a method by which to investigate the influence of the number of contact points on the stiffness coefficient. The relationship between displacement and force space is established via the proposed stiffness matrix. This work aims to identify the displacement of the object by detecting the contact force or to identify the contact force on the contact surface by recognizing the displacement of the object. The experimental results verify the correctness of the theoretical model.

The rest of this paper is organized as follows: The contact stiffness model is proposed in Section 2; the stiffness matrix of soft fingertip contact is established based on the variation in stiffness of the linear springs during contact. In Section 3, we simulated the contact between soft fingertip and object to verify the accuracy of the material parameters incorporated in the soft fingertip constitutive model. In Section 4, we researched the relationship between displacement and external force and offer a comparative analysis of soft fingertips with different sizes, demonstrating the feasibility of the proposed model. In Section 5, we came to the conclusion that the contact stiffness matrix of the fingertip and grasp stiffness matrix can be obtained and adopted to predict the micro-displacement state under given external forces.

## 2. Modeling Contact Stiffness of a Soft Fingertip

When two bodies are brought into contact, the real contact area will usually be smaller than the nominal contact area, but with differ in contact between two soft bodies or between a soft body and a rigid body. We can visualize the contact region as small areas where asperities from one solid are squeezed against asperities from another body. Moreover, we can visualize the contact region where asperities from one soft surface are squeezed against asperities from another body. The real contact area will usually be larger than the nominal contact area. The contact may be represented by an equivalent system with a reduced elastic modulus $E={\left[\left(1-\upsilon_{1}^{2}\right)/E_{1}+\left(1-\upsilon_{2}^{2}\right)/E_{2}\right]}^{-1}$, where $\upsilon_{1}$, $\upsilon_{2}$, and $E_{1}$, $E_{2}$ are the shear moduli and Young’s moduli of the two surfaces, respectively [33].

Soft contact occurs on the contact surface between the object and fingertip. To facilitate the model, we assume the contact surface contains a group of concentric circles, inspired by human fingerprints, with a large number of equivalent elastic point contacts on these curves. We establish a group of frames on the contact points and represent the stiffness of the fingertip as a set of passive linear springs.

As shown in Figure 1a, the soft contact model consists of contour curves and equivalent contact points located on each curve. We used $C_{n}$ to represent the *n*th curve. A soft contact is represented as nine linear springs, expressed as $L_{ij}$ (*I* = 1, 2, 3, *j* = 1, 2, 3). They are positioned at three uniformly distributed points on the edge of the contour line. The index *i* represents the number of each equivalent contact point, and the index *j* represents the number of the linear spring at each contact point.

A possible distribution of contact points is shown in Figure 1b. Each spring force can be represented as a vector $L_{ij}=[\begin{array}{cc} s_{i}{}_{j} & r_{i}\times s_{ij} \end{array}]$, which is along the spring direction. $r_{i}$ is a position vector from the contact frame to the equivalent contact point. The origin of the contact frame is at the center of the contact surface. The z-axis is normal to the contact surface in the coordinate frame, and the x- and y-axes are tangential to the surface. The contact point is located at the center of the circle. Assuming the contact frame of reference is at the center of the contact surface, the position vector can be represented as follows [34]:(1)$$r_{i}=[\begin{array}{ccc} a\cos(\varphi_{i}) & a\sin(\varphi_{i}) & 0 \end{array}],$$
where $\varphi_{i}$ is the angle between the vector of $i^{\mathrm{th}}$ contact point and horizontal axis; and a is the radius of the closed curve.

We define the screw matrix $L_{ij}$ at each contact point can be grouped as follows:(2)$$L_{ij}=[\begin{array}{cccccc} l_{ij} & m_{ij} & n_{ij} & p_{ij} & q_{ij} & r_{ij} \end{array}],$$
where $l_{ij}$, $m_{ij}$, and $n_{ij}$ are the translational parts, and $p_{ij}$, $q_{ij}$, and $r_{ij}$ are the rotational parts of the screw, respectively.

We choose three equivalent contacts on the curve with interval angle of 120°. The elastic coefficients of the nine springs can be represented by $k_{ij}$(i = 1, 2, 3; j = 1, 2, 3).

### Stiffness Matrix Analysis of Soft Fingertip Contact

The elastic model of soft fingertip contact is established based on the variation in stiffness of the linear springs during contact, with their vectors arranged into the matrix *K_g_*:(3)Kg=SKST
where S=[L11,L12,L13,L21…L33] presents the screw matrix on the contact surface.
(4)K=[Diag(k11,k12,⋯,k33)], K∈ℝn×n

The set of stiffness is placed on the diagonal to form a diagonal matrix K, where the element on the diagonal $k_{ij}$ represents the stiffness of the *i*^th^ linear spring at the *i*^th^ equivalent contact point. The Cartesian coordinate system is aligned with the direction of $l_{ij}$, projecting each stiffness onto the x-axis. The first diagonal element of the stiffness matrix has three nonzero terms, expressed as follows:(5)$$\sum_{i=1}^{3}\sum_{j=1}^{3}l_{ij}^{2}k_{ij}=l_{11}^{2}k_{11}+l_{21}^{2}k_{21}+l_{31}^{2}k_{31}.$$

These three terms along the x-axis can be written as three independent contacts: $k_{11}=k_{1x},k_{21}=k_{2x},k_{31}=k_{3x}$. We suppose that the spring stiffness of each point along the same direction is equal: $k_{1x}=k_{2x}=k_{3x}=k_{x}$. The number of equivalent contact points is expanded from 3 to m. Then, Equation (5) can be converted into the following:(6)$$\sum_{i=1}^{3}\sum_{j=1}^{3}l_{ij}^{2}k_{ij}=\sum_{i=1}^{m}k_{ix}=mk_{x}$$
(7)$$\sum_{i=1}^{3}\sum_{j=1}^{3}m_{ij}^{2}k_{ij}=\sum_{i=1}^{m}k_{iy}=mk_{y}$$
(8)$$\sum_{i=1}^{3}\sum_{j=1}^{3}n_{ij}^{2}k_{ij}=\sum_{i=1}^{m}k_{iz}=mk_{z}.$$

This method can simplify the other three diagonal elements:(9)$$\sum_{i=1}^{3}\sum_{j=1}^{3}p_{ij}^{2}k_{ij}=\sum_{i=1}^{m}\left[k_{z}\left(y_{i}^{2}\right)+k_{y}\left(z_{i}^{2}\right)\right]$$
(10)$$\sum_{i=1}^{3}\sum_{j=1}^{3}q_{ij}^{2}k_{ij}=\sum_{i=1}^{m}\left[k_{z}\left(x_{i}^{2}\right)+k_{x}\left(z_{i}^{2}\right)\right]$$
(11)$$\sum_{i=1}^{3}\sum_{j=1}^{3}r_{ij}^{2}k_{ij}=\sum_{i=1}^{m}\left[k_{x}\left(y_{i}^{2}\right)+k_{y}\left(x_{i}^{2}\right)\right],$$
where the position of the contact point is $\left(\begin{array}{ccc} x_{i} & y_{i} & z_{i} \end{array}\right)$. All these off-diagonal elements are zero. The contact stiffness matrix after simplification is as follows:(12)$$K_{g}=\left[\begin{array}{cccccc} 3k_{x} & & & & & \\ & 3k_{y} & & & & \\ & & 3k_{z} & & & \\ & & & k_{rx} & & \\ & & & & k_{ry} & \\ & & & & & k_{rz} \end{array}\right],$$
where the last three diagonal elements can be written as $k_{rx}=\sum_{i=1}^{3}\left[k_{z}\left(y_{i}^{2}\right)+k_{y}\left(z_{i}^{2}\right)\right]$, $k_{ry}=\sum_{i=1}^{3}\left[k_{z}\left(x_{i}^{2}\right)+k_{x}\left(z_{i}^{2}\right)\right]$, $k_{rz}=\sum_{i=1}^{3}\left[k_{x}\left(y_{i}^{2}\right)+k_{y}\left(x_{i}^{2}\right)\right]$, which represents rotational stiffness.

We assume a contact surface can be expressed as *n* concentric closed curves, and each closed curve has *m* equivalent contact points. The schematic diagram of soft fingertip contact is shown in Figure 1, where the *m* equivalent contact points are located on a closed curve with radius of $a_{j}$:(13)$$a_{j}=\sqrt{S_{j}/\pi },$$

Where $S_{j}$ is the area of the circle with radius $a_{j}$. The direction angle of the position vector is as follows:(14)φ=2π/m.

We can express the contact stiffness matrix of the soft fingertip K as follows:(15)$$K=\left[\begin{array}{cccccc} mnk_{x} & & & & & 0\\ & mnk_{y} & & & & \\ & & mnk_{z} & & & \\ & & & k_{rx} & & \\ & & & & k_{ry} & \\ 0 & & & & & k_{rz} \end{array}\right]$$
(16)$$k_{rx}=\frac{1}{mn}\sum_{j=1}^{n}\sum_{i=1}^{m}k_{z}{\left[a_{j}\sin\left(i\varphi \right)\right]}^{2}=\psi_{k_{rx}}k_{z}$$
(17)$$k_{ry}=\frac{1}{mn}\sum_{j=1}^{n}\sum_{i=1}^{m}k_{z}{\left[a_{j}\cos\left(i\varphi \right)\right]}^{2}=\psi_{k_{ry}}k_{z}$$
(18)$$k_{rz}=\frac{1}{mn}\sum_{j=1}^{n}\sum_{i=1}^{m}\left\{k_{z}{\left[a_{j}\sin\left(i\varphi \right)\right]}^{2}+k_{z}{\left[a_{j}\cos\left(i\varphi \right)\right]}^{2}\right\}=\psi_{k_{rx}}k_{z}+\psi_{k_{ry}}k_{z}$$
(19)$$\psi_{k_{rx}}=\frac{1}{mn}\sum_{j=1}^{n}\sum_{i=1}^{m}{\left[a_{j}\sin\left(i\varphi \right)\right]}^{2}$$
(20)$$\psi_{k_{ry}}=\frac{1}{mn}\sum_{j=1}^{n}\sum_{i=1}^{m}{\left[a_{j}\cos\left(i\varphi \right)\right]}^{2},$$
where $\psi_{k_{rx}}$ and $\psi_{k_{ry}}$ are the stiffness coefficients along the *x*-axis and *y*-axis direction, respectively.

Equation (15) provides the contact stiffness values along with three translational paths and three rotational movements. However, the conventional point contacts with friction provide translational stiffness without rotational stiffness.

The values of stiffness coefficients under different combinations of *m* and *n* are listed in Table 1 and Table 2. For an elastic element with a cross-sectional area of 1 cm^2^, the reference [35] provided the values of stiffness coefficients as follows:(21)$$\begin{array}{l} k_{x}=k_{y}=167\;N/\mathrm{cm},k_{z}=500\;N/\mathrm{cm}\\ k_{rx}=k_{ry}=42\;N/\mathrm{cm},k_{rz}=27\;N/\mathrm{cm} \end{array}$$

We compared our work with the data from reference [36] to verify the proposed model. This soft finger uses values of $k_{x}$, $k_{y}$, and $k_{z}$ from Equation (21). The corresponding $\psi_{rx}$ and $\psi_{ry}$ are shown in Table 1 and Table 2. The results $k_{rx}$, $k_{ry}$, $k_{rz}$ are as follows:(22)$$k_{rx}=k_{ry}=44.75\;N/\mathrm{cm},\;k_{rz}=29.89\;N/\mathrm{cm}$$

Obviously, the differences of $k_{rx}$, $k_{ry}$ and $k_{rz}$ between our model and reference [25] data are 6.5%, 6.5%, and 10.7%, respectively. The results from Table 1 and Table 2 were shown as follows: as the value of *n* increases, the error will be reduced; when n>3, the scale of decrease is less than 4%; we choose *m* = 3, *n* = 3.

## 3. Simulation of Contact Characteristics between Soft Fingertip and Object

The model of the soft fingertip and grasped object was constructed using the finite element analysis software Abaqus. The contact type between the fingertip and the object was a soft contact, whereby the fingertip model was simplified as a semicircular sphere, and the object was simplified as a rigid plate.

Figure 2 depicts the finite element model of a rectangular plate with the dimensions 0.5 mm × 30 mm × 25 mm alongside the finite element model of a semi-cylindrical finger end with a radius of 10 mm. Due to the metal plate being harder than the finger end, the deformation of the metal plate is negligible. The material of the plate is elastic steel, with a density of 7850 kg/m^3^, Poisson’s ratio of 0.3, Young’s modulus of 210,000 MPa, and C3D8R hexahedral elements. The plate is meshed with 600 units, each element with a length of 1 mm, a width of 1 mm, and a thickness of 0.5 mm. The material of soft fingertip is set as rubber, and the model type is hyper-elastic constitutive, with parameters set as follows: rubber density is 1.1 g/cm^3^. The finger end was meshed with C3D8RH and C3D6H elements, while the connection type between the soft finger and the object is static general nonlinear. The connection mode was defined as surface-to-surface, and the friction coefficient was set at 0.5, which is typically used for contact between silicone rubber and plastic materials. Additionally, all nodes on the upper surface of the fingertip are fixed in place.

To verify the accuracy of the material parameters incorporated in the soft finger constitutive model, a combined simulation and experimental testing of finger material characteristics was carried out. The simulation involved fixing the spherical bottom surface of the fingertip and applying vertical pressure on the object pressed into the fingertip. The relationship between the contact force and pressing distance is illustrated in Figure 3. The relationship curve derived from the simulation bears a resemblance to that from the experiment, with a relative error of under 3% within 3 mm of the pressing depth. This confirms the accuracy of the material parameters utilized in the finite element model.

As can be seen from the comparison diagram depicting the simulation and experimental curves, the finite element curve and experimental curve are essentially consistent when the deformation of the soft finger is less than 3 mm. However, when the deformation of the finger exceeds 3 mm, a deviation between the finite element curve and experimental curve arises. Moreover, the simulated contact pressure is greater than the pressure obtained via theoretical calculation at the same deformation. The difference between the two becomes increasingly larger as the deformation escalates. This outcome arises as the compression exceeds 1/3 of the total thickness, and the Hertzian model fails in agreement with reference [33]. As a result of this effect, more force is needed to produce a small amount of compression deformation in the soft finger ending. This force is also the reason behind the rapid increase in contact pressure when the deformation of the finger end exceeds 3 mm. Figure 4 shows the pressure cloud map of the pressing surface of the soft fingertip.

A comparative analysis of the simulation and experimental results was undertaken in order to evaluate the applicability of the proposed theory. The results showed that the theory has a certain range of application; specifically, when the deformation is less than 1/3 of the total thickness, the relationship between the contact force and the deformation of the fingertip can be accurately described. Figure 4 illustrates the accuracy of the theoretical predictions at various depths of deformation: (a) the depth of 1 mm; (b) the depth of 2 mm; and (c) the depth of 3 mm. These results provide valuable insights into the pressure distribution on the soft fingertip during forward pressing. The pressure distribution diagram in Figure 4a–c clearly demonstrates that the maximum contact force is located in the center of contact, with a subsequent decrease in contact force as one moves towards the edge, eventually reaching zero force at the contact edge. This phenomenon can be explained through the differential model of the soft fingertip, observed where the strain of the micro-element in the center of contact is the largest, resulting in the corresponding maximum contact pressure. Conversely, the micro-element at the contact edge does not compress and deform, resulting in zero contact pressure. These simulation results will aid in further understanding of contact mechanics and improve the development of soft fingertip.

## 4. Experimental Study and Discussion

The experimental setup, as shown in Figure 5, consists of two soft fingers and a 100 mm × 100 mm × 100 mm cube that is equipped with an MPU6050 gyroscope (InvenSense Inc, Sunnyvale, CA, USA). This gyroscope accurately records the angle and displacement of the cube within the fixed coordinate frame. Additionally, two linear motors are employed to generate thrust and pull force, while two sensors are utilized to measure the applied forces. The soft fingertips are skillfully crafted from silica gel. To apply an external force, F1 is exerted on the target at a vertex along the horizontal direction, which can be precisely controlled by the linear motor. Furthermore, F2 is applied vertically on the top of the cube to counteract the force of gravity.

In the experimental setup, one soft fingertip is attached to the end of the guide rod, while the other is securely fastened to the vertical plate. The target object is positioned between these two soft fingertips, with two contact points located at the center of the target’s left and right surfaces. To examine the influence of contact force, we selected force values of 20 N, 30 N, and 40 N. We recorded the object’s position as the external force progressively increased by 1 N. During these experiments, we utilized soft fingertips with diameters of 20 mm and 30 mm and elastic moduli of 1 MPa and 0.3 MPa, respectively.

The fabrication of the soft fingertip involves the use of a 3D-printed mold, as depicted in Figure 3. The fabrication process can be divided into two main parts. First, the chamber mold and the cover mold are designed using 3D software (SOLIDWORKS 2020). The chamber mold features a convex curved surface in the middle, which allows the silica gel to solidify and form an air cavity body that conforms to the shape of the mold. Similarly, the cover mold has a concave hemispherical shape. Both molds are then printed in a 3D printer. Second, the preparation of the fingertip involves mixing equal weights of the two-part silicone rubber parts, specifically parts A and B (Ecoflex0050 1A 1B), as recommended by the manufacturer (Smooth-On, Marcokey-USA). The mixed liquid silica gel is poured into the first cover mold, and then the second cover mold is assembled with the first one. Next, the assembled mold is placed in a vacuum box, and a vacuum pump (Mastercool 90066-2V-220, Randolph-USA) is used to reduce the pressure in the box from 0 to −0.09 Mpa. This process helps eliminate the small bubbles from the liquid silica gel that precipitate out of the material. After approximately 20 min, the assembled mold is removed from the vacuum box and transferred to an oven for curing at a temperature of 50 °C. The mold is left in the oven for 2 h until the silicone air cavity inside solidifies into a solid body. Finally, each sample is inspected using a directed light source to identify any remaining air bubbles within the material, as shown in Figure 6.

We utilize the PD method for controlling the linear motor to precisely generate pulling and thrust forces. In this experiment, our aim is to explore the correlation between applied force and object movement by comparing the theoretical analysis values with the experimental ones, as illustrated in Figure 7. We will study the relationship between force and displacement in soft finger grasping of rigid objects via four aspects: (1) degree of conformity between actual displacement and the theoretical model under varying external forces; (2) degree of conformity between actual displacement and the theoretical model under varying contact areas and external forces; (3) degree of conformity between actual displacement and the theoretical model under varying elastic moduli; (4) degree of conformity between actual displacement and the theoretical model under varying grasping forces.

### 4.1. Degree of Conformity between Actual Displacement and the Theoretical Model under Varying External Forces

In this particular experiment, a soft fingertip with a diameter of 20 mm and an elastic modulus of 1 MPa was used. The contact force for the experiment was set at 30 N. The experimental setup and process can be observed in Figure 8.

Table 3 displays the extracted pixel coordinates of P_1_, P_2_, and P_3_ for each image. These pixel coordinates are obtained by detecting the spatial orientation of the target object, as illustrated in Figure 9. This process allows us to determine the coordinates of the target object’s vertex in the camera coordinate system.

Figure 10 depicts the three orientational components, with x, y, and z representing the displacements. The figures also include theoretical curves to facilitate comparison with the experimental results. Notably, since the directions of the external forces are perpendicular to the y- and z-axes, and the stiffness values are equal in y and z directions, leading to the theoretical curves for y and z being overlapped. The displacement curves for each component exhibit a linear trend and demonstrate a direct correlation with the magnitude of the applied external force, consistent with theoretical expectations. The x and y components of the displacement closely align with the theoretical curves in terms of slope, while the z component deviates slightly below the theoretical curve. However, the absolute slope value is greater than that which was predicted by the theoretical model. This discrepancy can be attributed to a slight positive shift along the z-axis, resulting from the positional deviation of the contact point from the surface’s center.

### 4.2. Degree of Conformity between Actual Displacement and the Theoretical Model under Varying Contact Areas and External Forces

In this experiment, fingertips of varying sizes were used along with different contact forces. The modulus of elasticity was set at 1 MPa. The experimental procedure is depicted in Figure 11.

As shown in Figure 12, the x and y components of the displacement screw are linear for 20 mm- and 30 mm-sized fingertips, consistent with theoretical analysis. The slopes of the two curves are almost the same, consistent with that of theory work. In addition, θx,θy,θz are also basically linear. Moreover, the θx component obtained in the experiment is lower than that of the theoretical value, which means that the angle about the *y*-axis was larger than that of the theoretical value. The most likely reason is a slight shift in the grasping position by the soft fingertips.

### 4.3. Degree of Conformity between Actual Displacement and the Theoretical Model under Varying Elastic Moduli

In this experiment, the fingertip was analyzed under different elastic moduli, as shown in Figure 13. The experimental results are shown in Figure 14. The y and $\theta_{x}$ components of displacement screw are both zero, regardless of whether the soft fingertips’ elastic modulus is 1 MPa or 0.3 Mpa, which is consistent with the experimental results. For soft fingertips with the smaller elastic modulus, the $x,\theta_{y},\theta_{z}$ components of the displacement screw and the absolute values of the theoretical curves’ slopes are greater than those of fingertips with the larger elastic modulus, which is also consistent with the experiment. Thus, soft fingertips with a larger elastic modulus provide more substantial constraints on the target object; hence, its orientation is less likely to change upon the application of external force.

### 4.4. Degree of Conformity between Actual Displacement and the Theoretical Model under Varying Grasping Forces

In this experiment, the grasping forces with varied magnitudes were presented. In Figure 15a,b, the *x, θy,* and *θz* components of the displacement screw exhibit a linear dependence on external force, albeit with experimental error. Note that if an experimental curve matched its theoretical counterpart, the *y* and *θx* components were excluded to facilitate comparison. It is apparent that as the contact force decreases, the trend between displacement and the external force becomes steeper, which aligns with both theory and intuition. With the same external force, a smaller grasp force yields a more substantial change in the displacement screw, indicating weaker constraint by the soft fingertips on the object.

An exception arises in Figure 15 for the *θy* and *θz* components of displacement, where the absolute value of the experimental curve slope is slightly lower than that of the theoretical curve at 40 N of contact force. This discrepancy may be attributed to greater fingertip deformation when contact force is higher. Additionally, the incompressibility of materials may cause the axial and tangential stiffness of a fingertip to exceed theoretical predictions.

## 5. Conclusions

The manipulation of robotic hands with soft fingertips has become increasingly important in recent years, with applications ranging from manufacturing to healthcare. However, the effective control of a multi-fingertip robot is dependent on understanding the nature and modeling of soft fingertips during grasping and manipulation. Soft fingertips are a critical component of robotic hands as they enable the hand to interact with objects in a manner similar to human hands. Understanding the contact mechanics of soft fingertips is necessary in order to design effective robotic hands that can perform a wide range of tasks. The development of effective robotic hand control requires a comprehensive understanding of the contact mechanics of soft fingertips. The proposed contact stiffness model and six-dimensional representation method provide a mathematical model for the contact stiffness of soft fingertips, making it possible to analyze and predict the behavior of soft fingertips during grasping and manipulation. The experimental results have demonstrated the feasibility of the proposed model, providing a foundation for the development of effective robotic hand control with soft fingertips.

In this work, we have presented a mathematical formula based on the Winkler Model and a six-dimensional representation method for soft fingertip contact. The proposed model provides a mathematical model for the contact stiffness of soft fingertips, making it possible to analyze and predict the behavior of soft fingertips during grasp and manipulation. The six-dimensional representation method considers the three-dimensional forces and displacements in the x, y, and z directions, as well as the three-dimensional moments and rotations about the x-, y-, and z-axes. This comprehensive approach provides a more accurate representation of the behavior of soft fingertips during grasping and manipulation. To verify the proposed contact stiffness model, we analyzed and calculated several soft fingertip grasping examples. Our research has shown that the soft multi-fingered hand can effectively manipulate the attitude and position of objects by controlling the stiffness of fingertips and adjusting the contact force. By utilizing the contact stiffness matrix of the fingertip and grasp stiffness matrix, it is possible to predict the micro-displacement state when subjected to specific external forces. Experimental results have demonstrated that the position error between theoretical work and tested data was less than 9% along the x, y, and z directions, while the angle error was less than 5.58% about the x-, y-, and z-axes. This coherence between experiments and simulations proves the feasibility of the proposed contact stiffness model for robotic hands with soft fingertips.

The limitation of this research is that the experiment was conducted using a flat contact surface rather than a rough plane with varying shapes. This direction can be further explored. The displacement state under different grasping forces and external forces will vary when the shape of the fingertip differs. The state of the object undergoes significant changes with variations in the angles of the contact surface. Research on the unified mathematical expression of contact stiffness for differently shaped fingertips is crucial for the grasping and manipulation of dexterous hands and has great practical value. In our future work, we plan to enhance the experimental device by incorporating precise force measurement sensors and accurate angle measurement sensors. This improvement aims to reduce measurement errors and introduce a contact angle adjustment mechanism between fingers and objects. Ultimately, our goal is to achieve precise control of objects using soft fingers.

## Figures and Tables

**Figure 1 biomimetics-08-00398-f001:**
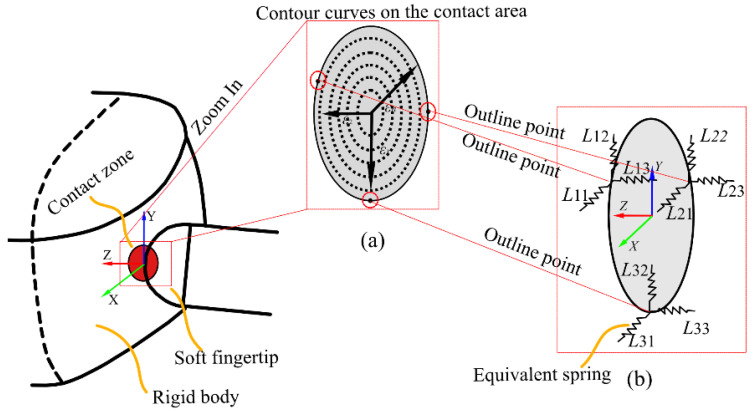
Equivalent representation of soft fingertip contact. (**a**) The contour curves on the contact area. (**b**) The contact equivalent points on the contour curves.

**Figure 2 biomimetics-08-00398-f002:**
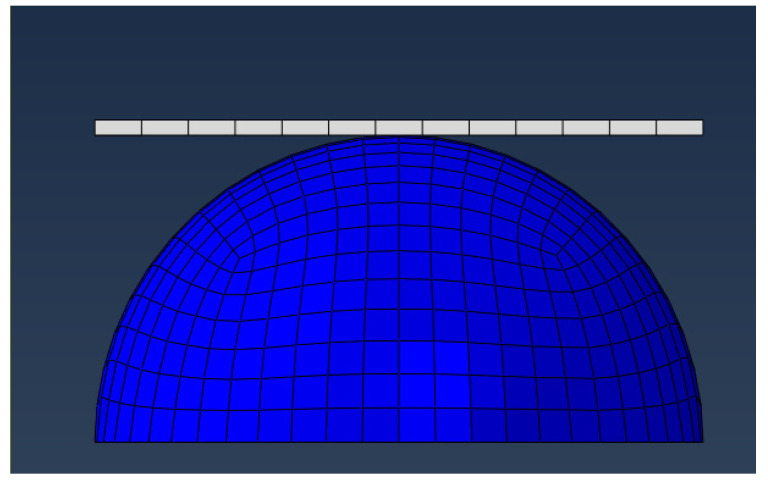
The meshing of soft fingertip model.

**Figure 3 biomimetics-08-00398-f003:**
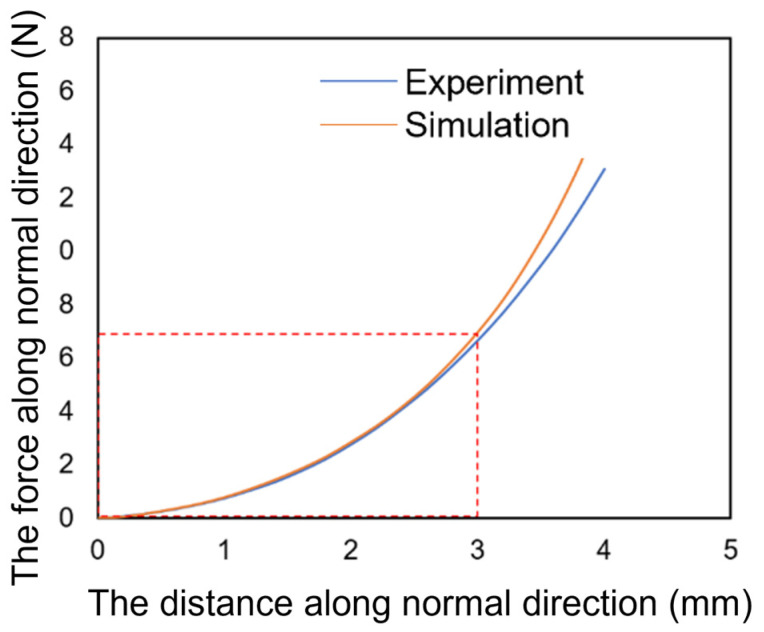
The normal force and normal compression displacement curves.

**Figure 4 biomimetics-08-00398-f004:**
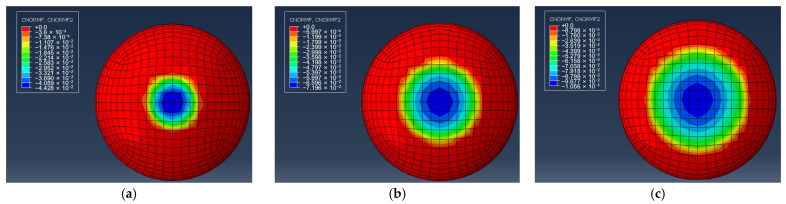
The cloud images depict the pressure distribution on the contact surface at various depths of deformation: (**a**) The depth of deformation d_0_ = 1 mm; (**b**) The depth of deformation d_0_ = 2 mm; (**c**) The depth of deformation d_0_ = 3 mm.

**Figure 5 biomimetics-08-00398-f005:**
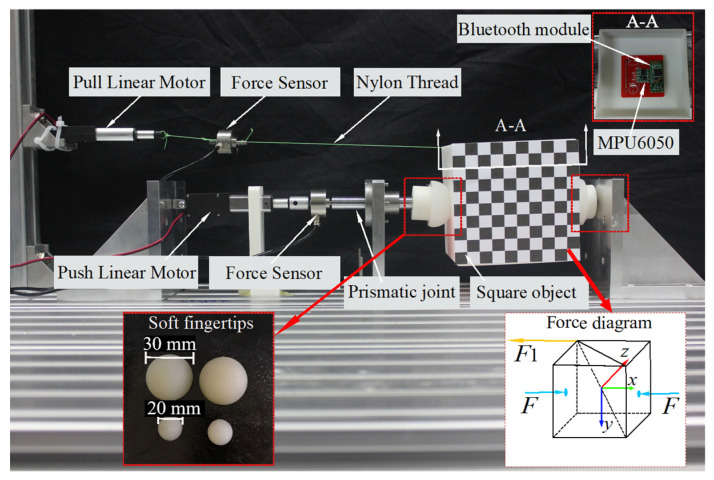
Experimental setup.

**Figure 6 biomimetics-08-00398-f006:**
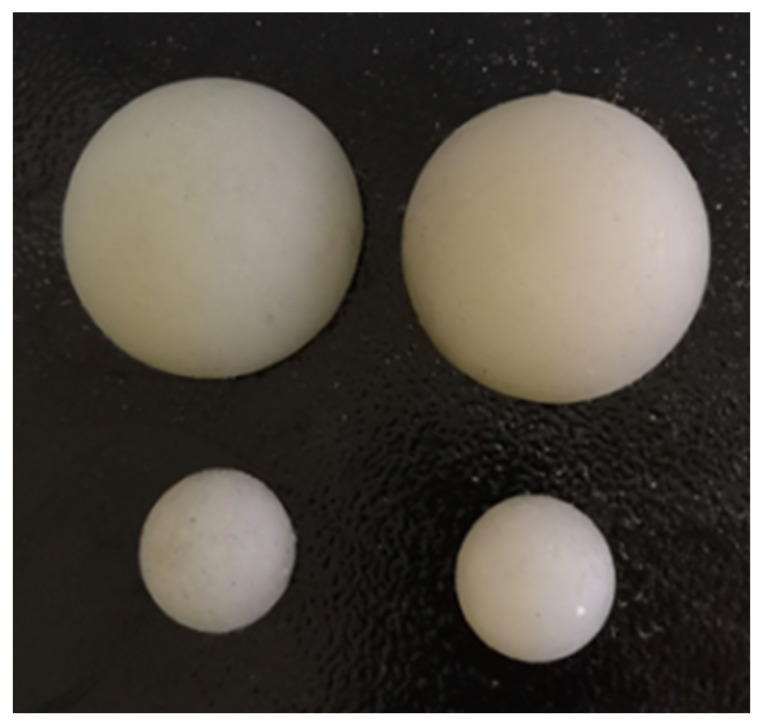
Soft fingertip prototypes with different sizes.

**Figure 7 biomimetics-08-00398-f007:**
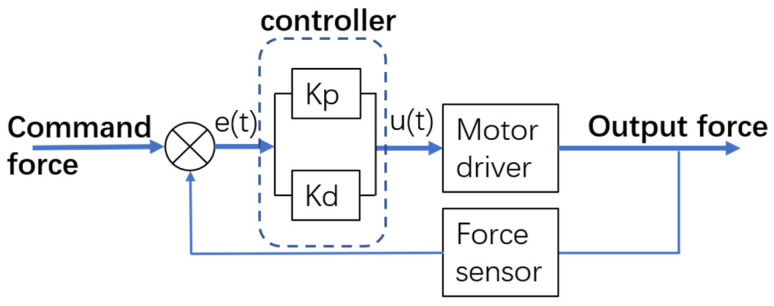
PD control block diagram of linear motor.

**Figure 8 biomimetics-08-00398-f008:**
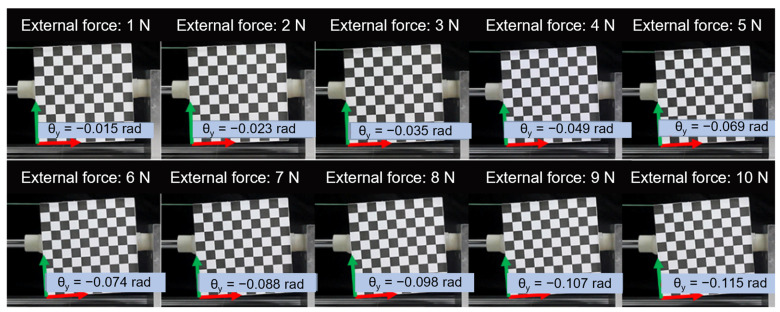
Photos of target object orientations under various external forces.

**Figure 9 biomimetics-08-00398-f009:**
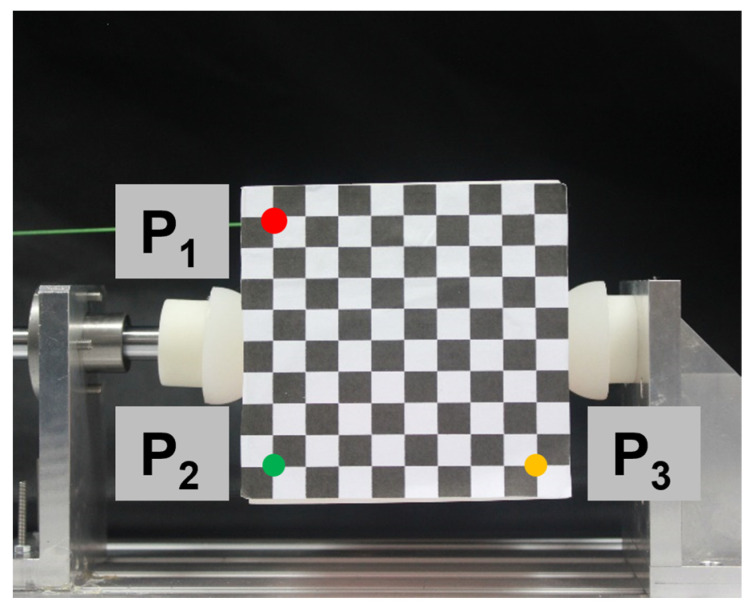
Calculation of the target displacement.

**Figure 10 biomimetics-08-00398-f010:**
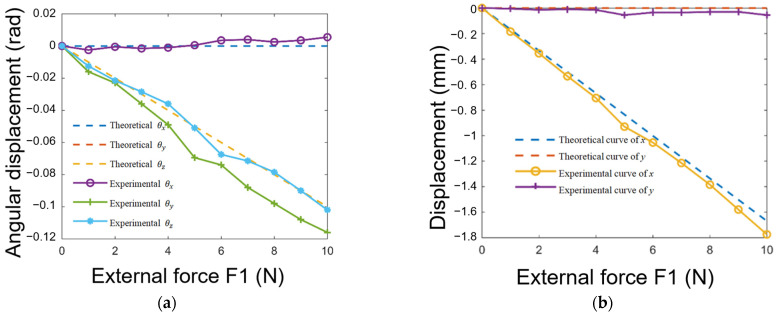
Effects of varied forces on object angle and displacement: (**a**) variations in relative angular displacement $\theta_{x},\theta_{y},\theta_{z}$ as a function of external forces; (**b**) variations in relative translational displacement along the x-, y-, and z-axes as a function of external forces.

**Figure 11 biomimetics-08-00398-f011:**
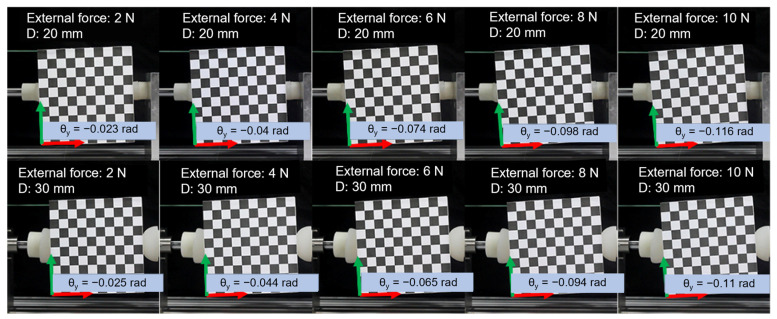
Photos of target object orientations under different forces applied with different sizes of fingertips. The modulus of elasticity was 1 MPa, the diameters of the soft fingertips were 30 mm and 20 mm, respectively, and the external forces were 2 N, 4 N, 6 N, 8 N, and 10 N, respectively.

**Figure 12 biomimetics-08-00398-f012:**
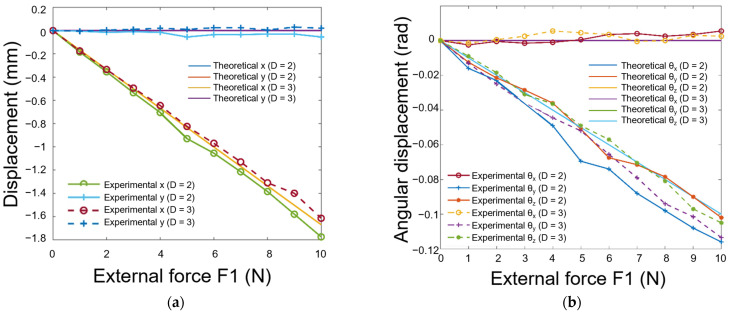
Comparative analysis of different forces applied to an object using two different fingertip sizes (D = 2 means the diameter is 20 mm; D = 3 means the diameter is 30 mm): (**a**) the relative displacement changes versus external forces; (**b**) the relative angular displacement changes versus external forces.

**Figure 13 biomimetics-08-00398-f013:**
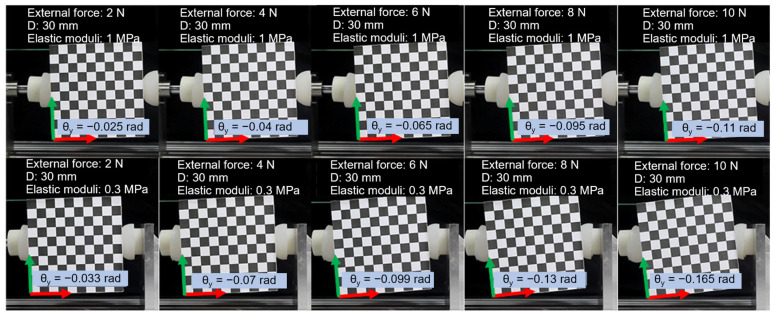
Photos of target object orientations under different elasticity moduli of 1 MPa and 0.3 MPa, both with diameter of 30 mm and grasp force of 30 N. The external force applied ranged from 2 N to 10 N in increments of 2 N.

**Figure 14 biomimetics-08-00398-f014:**
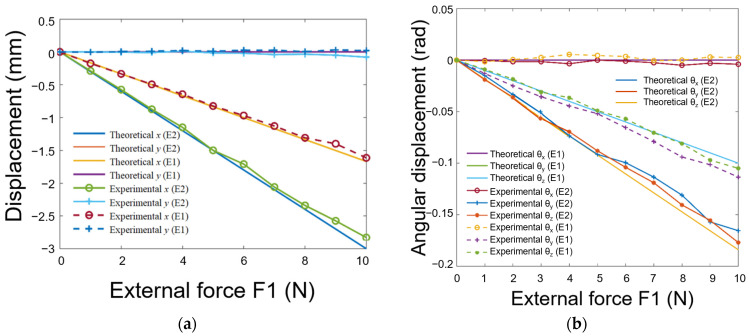
The object was subjected to different forces using two distinct elasticity modulus fingertips (E1 = 1 MPa; E2 = 0.3 MPa), resulting in the following: (**a**) the relative changes in translational displacement plotted against external forces; (**b**) the relative changes in angular displacement plotted against external forces.

**Figure 15 biomimetics-08-00398-f015:**
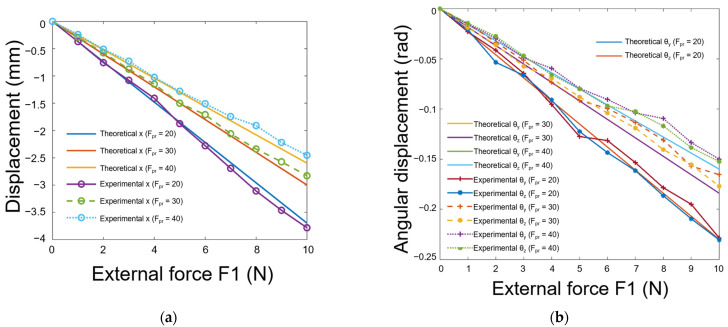
The object was subjected to varying grasping forces (F_pr_ = 20, 30, and 40 N), resulting in the following: (**a**) the translational component of the displacement screw plotted against external force for each grasping force; (**b**) the rotational component of the displacement screw plotted against external force for each grasping force.

**Table 1 biomimetics-08-00398-t001:** The values of $\psi_{rx}$, $\psi_{ry}$ for different combinations of m and *n*.

Variables	*n* = 1	*n* = 2	*n* = 3	*n* = 4	*n* = 5	*n* = 6	*n* = 7	*n* = 8
m = 3, 4, 5, 6	0.159	0.119	0.106	0.099	0.095	0.092	0.090	0.089

**Table 2 biomimetics-08-00398-t002:** The value of $\psi_{rz}$, for different combinations of m and *n*.

Variables	*n* = 1	*n* = 2	*n* = 3	*n* = 4	*n* = 5	*n* = 6	*n* = 7	*n* = 8
m = 3, 4, 5, 6	0.318	0.238	0.212	0.199	0.191	0.185	0.181	0.179

**Table 3 biomimetics-08-00398-t003:** Pixel coordinates of P_1_, P_2_, and P_3_ with different external forces.

External Force *F*_1_ (N)	Coordinate of P_1_		Coordinate of P_2_		Coordinate of P_3_	
	x	y	x	y	x	y
0	479	233	484	669	922	663
1	479	233	485	669	923	662
2	479	234	487	669	924	661
3	479	234	488	670	926	660
4	479	235	490	670	927	659
5	479	235	491	671	929	658
6	479	235	493	671	931	657
7	480	236	495	671	932	655
8	480	236	497	672	934	654
9	480	236	499	672	936	652
10	480	237	500	672	938	651

## Data Availability

Data are available from the authors upon request.

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
