# Peer review of "Modeling Contact Stiffness of Soft Fingertips for Grasping Applications"

_biomimetics, 2023, doi:10.3390/biomimetics8050398_

Round 1

Reviewer 1 Report (Previous Reviewer 1)

The authors tried to answer the questions raised. Some methodology problems remain, but the reviewer acknowledges the efforts and work of the authors.

Reviewer 2 Report (Previous Reviewer 2)

The paper could be accepted in this last form.

This manuscript is a resubmission of an earlier submission. The following is a list of the peer review reports and author responses from that submission.

Round 1

Reviewer 1 Report

The manuscript works on contact modeling between soft fingertips and rigid objects, considering a simple Winkler model at three contact points.

The considered assumption of no sliding motion condition at tangential directional is restrictive, especially for soft fingertip models. Moreover, hyperelastic model should be used to simulate soft materials instead of elastic model.

The iterative nonlinear method used in the manuscript is not detailed. The considered Winkler model is not suitable to model 3D frictional contact. Certainly, the soft fingertips contact does not presents constant stiffness based only at three discrete points. Langrange multiplier will be more suitable for this kind of modelling. The assumption that the spring stiffness of each point is equal for all directions implies in tangential force equal to normal force. This condition can be obtained for very specific contact conditions (for instance, dry contact between rubber and steel). Results do not prove that the use of silica gel on soft fingertips works for the considered contact condition. More details are necessary to reproduce numerically or experimentally the presented results.

A sensibility analysis of the “series of frames on the contact points” must be performed. Only three “equivalent” contact points seems to be not enough. The use of two soft fingertips, with diameters of 20mm and 30mm, and elastic moduli of 1 MPa and 0.3 MPa are also not justified. Why authors choose these values? It seems to be an empirical decision.

Theoretical angular displacement for Y direction is not visible in Figure 7a. Also, authors should check if all responses are presented in Figures 9, 11 and 12.

Some parts of the text are self-indulgent, for instance: “we meticulously recorded…” and “mixed liquid silica gel is carefully poured…”.

At the end of section 3, authors comment about the incompressibility of materials, but the definition of materials are not well established. Despite promising results for a narrow range of applied force, materials and methods section must be completely redone.

Among thirty seven presented references only five works are from last five years. Some references presented missing information and format problems (authors should followed journal template).

In short, the manuscript presents an interesting application but state-of-the-art revision, methodology and results discussion must be fully reviewed before it can be considered for publication.

The text is clear. The English language quality of any manuscript can be improved.

Author Response

Thank you for the kind work on our manuscript. The reviewers provided constructive comments. We are grateful for having the opportunity to respond to them in a revised version.

Reviewer 2 Report

The authors in the conclusion must  propose some future research to decrease the position error between theoretical work and tested data to be less than 9% and the angle error to be less than 5.58%.

Author Response

(The authors gave the same response as above.)
